# Near-IR & Mid-IR Silicon Photonics Modulators

**DOI:** 10.3390/s22249620

**Published:** 2022-12-08

**Authors:** Georgi V. Georgiev, Wei Cao, Weiwei Zhang, Li Ke, David J. Thomson, Graham T. Reed, Milos Nedeljkovic, Goran Z. Mashanovich

**Affiliations:** Optoelectronics Research Centre, University of Southampton, Southampton SO17 1BJ, UK

**Keywords:** modulators, silicon photonics, depletion, PAM-4, germanium

## Abstract

As the silicon photonics field matures and a data-hungry future looms ahead, new technologies are required to keep up pace with the increase in capacity demand. In this paper, we review current developments in the near-IR and mid-IR group IV photonic modulators that show promising performance. We analyse recent trends in optical and electrical co-integration of modulators and drivers enabling modulation data rates of 112 GBaud in the near infrared. We then describe new developments in short wave infrared spectrum modulators such as employing more spectrally efficient PAM-4 coding schemes for modulations up to 40 GBaud. Finally, we review recent results at the mid infrared spectrum and application of the thermo-optic effect for modulation as well as the emergence of new platforms based on germanium to tackle the challenges of modulating light in the long wave infrared spectrum up to 10.7 μm with data rates of 225 MBaud.

## 1. Introduction

Silicon photonics has been developed as one of the most competitive solutions for integrated photonics towards high bandwidth interconnects beyond 100 GBaud in data centers, low power consumption and high density integrated co-packaged optics for future data communications. The optical modulator is a critical component in these systems, not only in traditional data communication links but also in microwave photonics or chip-scale computing networks. Since the breakthrough work reported by Intel in 2004 [1], the silicon modulator has become one of the hottest research topics in silicon photonics. A thorough review of its development over the past two decades can be found in [2,3].

In this paper, we review recent developments in the near-infrared (near-IR) and mid-infrared (mid-IR) group IV photonic modulators that show promising performance. Our objective for this review is to:Highlight state of the art results.Focus on new research trends associated with group IV modulators, such as the convergence of photonics and electronics in the C band, the implementation of advanced modulation techniques in the 2 μm band and recent emergence of germanium as the platform of choice for mid-IR modulators.Discuss areas for further research and improvement.

To allow the reader to compare devices, we conduct the performance review along the following figures of merit:Insertion loss (IL)—measured in dB; the total optical loss increase by inserting the device into the optical path.Modulation speed—measured in Baud; the maximum symbol rate at which the modulator can transmit information.Extinction ratio (ER)—measured in dB; the optical intensity of the ‘1’ signal divided by the optical intensity of the ‘0’ signal, also known as modulation depth.Modulation efficiency ($V_{\pi }L_{\pi }$)—measured in V·cm; the voltage required for a π phase shift in a given length.

High-speed silicon modulators with a speed of 112 GBaud PAM-4 have been demonstrated by co-design of CMOS electronics and photonics Mach–Zehnder modulators (MZM) [4], and 120 GBaud PAM-4 has also been realised in compact silicon microring modulators (MRM) [5]. The MRM version modulators benefit from compact sizes < 10 μm and low capacitance, allowing high bandwidth densities and high energy efficiencies in terabit links now [6] and petabit scale links in future [7]. In the 2 μm band compact MZM have been realised performing at 12.5 GBaud PAM-4 [8] and 40 GBaud PAM-4 [9]. The mid-IR modulators have been shown to operate at 60 MBaud at 3.8 μm in Germanium-on-silicon [10] and 112 MBaud at 10.7 μm in graded silicon-germanium waveguides [11]. In this paper we review the state of the art of near-IR and mid-IR group IV modulators.

## 2. Near-IR Group-IV Modulators

Silicon-based MZMs have energy consumption at a level of a few pJ/bits. A π phase shift change requires phase shifter lengths typically of at least several millimetres and driver voltages above 5 V. The carrier induced absorption for a π phase shift change is high about 10 dB, which intrinsically cannot be eliminated. To pursue better performing modulators in silicon photonics, new material integration with Si, such as LiNbO_3_ [12,13], BTO [14], PZT [15] and EO polymers that have the linear optoelectrical effect [16,17] have been demonstrated with lower insertion losses. LiNbO${}_{3}$ modulators show much higher EO modulation bandwidths (100 GHz) compared to monolithic silicon modulators and their performance can be engineered with high speed electrode design [18]. BTO modulators show a Pockels coefficient of as high as 923 pm/V [14], which is the highest efficiency demonstrated in silicon photonics for hybrid inorganic materials. In addition to growing BTO films with MBE [13,14], RF sputtering can also be used in heterogeneous BTO-SiN modulators with an equivalent Pockels coefficient of ≈160 pm/V [19]. The currently demonstrated BTO-based modulators show bandwidth up to 30 GHz [20]. EO polymers also show very high EO modulation coefficients $n^{3}r$ > 2000 pm/V [16], and their demonstrated modulator features are highly competitive with other inorganic modulators. The demonstrated modulators have $V_{\pi }=$ 1.5 V, insertion loss <1 dB and bandwidth of 40 GHz for a compact phase shift length of 280 μm [21]. However, their CMOS compatibility still has to be addressed as it potentially hinders their applications and raises co-integration challenges with lasers and detectors [22].

In terms of semiconductor solutions, n-type III-V/Si MOS modulators were proposed to achieve the highest modulation efficiency of $V_{\pi }L_{\pi }$ = 0.047 V·cm [23] by taking advantages of their high electron mobility, low carrier-plasma absorption and large electron-induced refractive-index change. However, their RC bandwidth is limited to less than 1 GHz. A further compromise of the modulation efficiency and bandwidth has to be adopted to reach data rates of 12.5 GBaud and $V_{\pi }L_{\pi }$ of 0.37 V·cm [24]. So far, the highest data rate of III-V/silicon hybrid MOS modulators has been about 25 GBaud at a modulation efficiency of 1.3 V·cm [25]. High-speed operation is achieved by drastically reducing the modulation efficiency or by reducing the loaded capacitor. In terms of bandwidth, hybrid III-V/silicon modulators are not superior to silicon-insulator-silicon capacitor (SISCAP) modulators [26]. Hence, the trade-off between modulation bandwidth and efficiency for III-V/Si structures points to other physical mechanisms such as Franz–Keldysh carrier depletion effect in reverse bias regime [27]. The RC bandwidth is estimated to be above 100 GHz benefiting from a small depletion capacitance. In such a scenario, carrier accumulation regime is not efficient enough for capacitor modulators as it requires loading much higher capacitance and, therefore, high-speed operation cannot be achieved. For other types of capacitor modulators, such as the single layer graphene/insulator/Si capacitor, modulation efficiency of 0.28 V·cm has been achieved with suppressed optical losses by Pauli blocking. The demonstrated modulation bandwidth is currently limited to 5 GHz [28]. With an optimised trade-off between modulation bandwidth and capacitance, the current MOS modulator development in 220 nm SOI platform shows that 50 GBaud PAM-4 modulation can be achieved for sub-mm size MZI modulators [29] and 100 GBaud OOK modulation for ring modulators [30], by using lateral silicon/SiO${}_{2}$/polysilicon junctions.

In one experimental demonstration, SOI rib waveguides have been shown to suffer from two photon absorption(TPA) and surface carrier absorption for guided mode powers greater than 20 dBm [31] In a typical test system, the tunable laser power is 13 dBm and a grating coupler with approximately 5 dB loss is used to couple the light into a single mode waveguide. Furthermore, in a balanced MZI modulator, the light is split into two waveguide arms for a further 3 dB power reduction. Therefore, the TPA effects can be neglected for standard PN MZI modulators. The TPA effect is more pronounced in ring resonators because of the higher electric field intensity caused by the resonance, and results in a change of the dynamic response of the resonators at high optical power input. It has been proved in [32], however, that despite the effect of TPA the dynamic response of ring modulators can be controlled well using electronics circuits for 100 Gb/s PAM-4 operations.

For ultra-compact modulation, GeSi and Ge/SiGe quantum well (QW) structures formed waveguide electro-absorption modulators (EAM) with a 3 dB bandwidth of 50 GHz, small footprint (<500 μm), low energy consumption 100 fJ/bit, covering the O, C and L communication bands [33,34,35]. The integration of these EAMs requires complex epitaxial growth and material engineering [36,37,38]. To cover different optical bands, the GeSi composition ratios have to be varied [33,34,35,38,39]. Ge/SiGe quantum well (QW) structures are more complicated and require engineering strain relaxed buffer layers, strain balanced QWs and maximising the electrical field in QWs [37]. The absorption of EAMs is usually narrowed down to small optical bandwidths (20–50 nm) and shows strong wavelength dependent figure of merit, defined as extinction ratio over insertion loss (ER/IL). In comparison with GeSi and Ge/SiGe-QW EAMs, silicon micro ring resonator modulators have also been demonstrated with a compact footprint (diameter 5 μm) [40], low insertion loss and high extinction ratio from a low drive voltage, low capacitance, low energy consumption <100 fJ/bit and with high bandwidth of 50 GHz [41]. The main drawback is that the operation is limited to discrete resonances separated with a resonator free spectral range, whose wavelength is sensitive to temperature variation and fabrication tolerances. Current works show that smart control of the ring resonators can enable ring modulators to operate with a data rate of 56 GBaud at PAM-4 formats [32]. Resonance shifts can also be co-adjusted with laser wavelengths. Hence, the operation stability will not become a critical problem in a practical application. Amplitude modulation has also been demonstrated for coherent I/Q modulation and high order modulation formats [42].

The integration of photonics and electronics was originally referred to as the physical coupling approach between photonics and electronics devices, such as wire-bonding [43,44], flip-chip bonding [45,46] and monolithic integration [47,48,49]. These approaches were commonly discussed and compared in terms of fabrication costs, yield, power efficiency and throughput. Previously, monolithic integration was considered a better approach regarding power efficiency and throughput, as it avoids additional parasitic effects associated with the device’s packaging. However, challenges within device fabrication and concerns about the yield are always associated with the monolithic solution. In contrast, the wire-bonding and flip-chip bonding approaches offer low costs solutions but introduce additional parasitic effects within the EO interface.

Regardless of the physical coupling approach adopted, the achievable baud rate for an EO integrated silicon transmitter was limited to a range of 56–64 GBaud/wavelength/ polarisation prior to 2020 if digital signal process (DSP) techniques were not adopted. Apparently, DSP techniques could significantly compensate for bandwidth limitation within the silicon modulators and the bandwidth drop within EO integration. Several works [5,43,50,51] have reported more than 100 GBaud transmission by using complex DSP techniques. The adoption of a complex modulation format could easily boost the overall throughput to more than 115 GBaud PAM-8 [52] for intensity transmission or 69 GBaud for DP-16QAM [46] coherent transmission.

DSP techniques are essential for optical communication links, but bottlenecks of silicon modulators (particularly the bandwidth limitation) require a further evolution of EO integration, which is the co-design and co-packaging of photonics and electronics. That means neither the electronics devices (i.e., driver amplifier) nor the photonics devices (i.e., modulator) can be treated as standalone components. Instead, the realisation of such a system requires a thorough understanding of photonics devices implementation, electronics circuit design and precise modelling of the parasitic effects that arise from device packaging.

In our recent work [53], we presented an example where a carrier depletion modulator is co-designed with a CMOS driver and can operate up to 100 GBaud (OOK) without using any DSP equalisations. Compared to most other works in this field, where the driver amplifier and modulator were modelled and designed separately, we have introduced a T-coil peaking network within the interface of the driver amplifier and silicon modulator. As highlighted in Figure 1a–c, this T-coil network is modelled by considering the depletion mode MZM as a load, whose net carrier concentration profile was modelled using Silvaco-TCAD. Once the dimensions of the T-coil network are determined, the CMOS driver could be implemented via standard analogue IC design flows, with its core-circuit diagram shown in Figure 1d. In order to facilitate device packaging requirements, the standard carrier depletion-based optical modulator is designed in a U-shape MZM configuration allowing access to both input and termination pads of the MZM on one side of the chip. This enables both ends of the modulator to be electrically connected to the CMOS driver by a flip-chip bonding process, which is shown in Figure 1e. A microscope view of the fully packaged device is shown in Figure 1f. The measurement results from a test structure show a modulation efficiency of 1.5 V·cm and a phase shifter loss of 2.7 dB/mm with a 1 V reverse bias voltage. The overall phase shifter length in this design is chosen to be 2.47 mm, which gives a total optical loss of 6.9 dB, including the losses from two MMIs, the passive waveguides and the 2.47 mm long phase shifters.

The performance of the co-packaged all-silicon transmitter is thoroughly analysed and compared with a standard 2 mm long silicon MZM, which is fabricated within the same SOI wafer. Firstly, the EO responses of both devices are tested, as shown in Figure 2a,b. As expected, the 3 dB EO bandwidths of the co-packaged silicon transmitters are considerably higher than the standalone devices. Highlighting the EO response at 67 GHz for comparison, the co-packaged devices are at −4.5 dB, whereas the standalone devices are at −8.2 dB. This means that co-design of the electronics and photonics devices could significantly alleviate the bandwidth limitation concerns of silicon modulators. Furthermore, when operating at the quadrature point, optical eye diagrams at 56 GBaud OOK, 80 GBaud OOK and 100 GBaud OOK, shown in Figure 2c–e, were detected, where no equalisation techniques were used. When a 7 tap feed-forward equalisation was adopted within the DCA, the co-packaged all-silicon transmitter could operate up to 118 G OOK while the CMOS driver only consumed 205 mW shown in Figure 2f.

In Table 1, we provide a summary of the demonstrated performances of state of the art near-IR modulators.

## 3. Mid-IR Group-IV Modulators

The mid-IR wavelength range (2–20 μm) is of particular interest for applications in environmental and biochemical sensing, medicine and communications. This range contains the fundamental transition bands for a multitude of gases and thus is useful for spectroscopic measurements [56,57]. In addition, many biological and chemical molecules have unique and strong absorption profiles in the mid-IR spectrum which allows for their detection [58,59,60]. For optical communications it presents an opportunity to expand the present near-IR spectrum into the mid-IR, enabling more optical channels and, hence, higher data throughput. The mid-IR also features two atmospheric windows (3–5 and 8–14 μm) offering additional spectrum for free-space transmission. Research groups have developed optical modulators operating at multiple wavelengths in the mid-IR from 2 μm to 11 μm.

The theoretical capacity limit of single mode fibres (SMF) is being approached due to techniques such as coherent detection and wavelength division multiplexing (WDM) [61]. As an even more data-hungry future looms ahead, capacity scaling of the fibre systems is necessary. Adding parallel systems to current infrastructure will be impractical in terms of cost and sustainability. This raises the possibility of looking at wavelength windows beyond the traditional telecomm band. One promising solution to this ’capacity crunch’ is the emergence of the 2 μm wavelength window for free space and fiber communications to act as a supplement to the existing 1550 nm C-band infrastructure [62]. Developments in hollow core photonic bandgap fibres (HC-PBGF) have shown lower loss compared to SMF [63], low latency [64] and high thermal stability in the 2 μm band [65]. Conveniently, this spectrum also overlaps with the gain window of thulium ions which enables the use of Thulium-doped fibre amplifiers (TDFA) with broad gain spectrum [66]. Furthermore, similarities in silicon photonics mode profiles between 2 μm and the current telecomms band allow for compatibility with already established semiconductor foundries. This has led to extensive interest in the development of electro-optical modulators (EOM) operating in the 2 μm wavelength window.

Modulation will also be required for some of the proposed longer wavelength sensing applications: in bulk absorption spectroscopy systems modulation is used in various schemes for improving the signal to noise ratio of optical absorption measurements. For example, free-space optical choppers are frequently used in synchronous detection schemes to reduce the measurement bandwidth and shift the carrier to a higher frequency with reduced 1/f noise, and phase modulators are a key component in frequency modulation spectroscopy systems [67] that can perform extremely high precision absorption measurements. Integration of fast, efficient, mid-IR modulators into photonic integrated circuits (PICs) will create exciting new opportunities for high performance but low cost absorption spectroscopy systems.

Among the different platforms used to develop modulators in the mid-IR, Silicon-on-insulator (SOI) remains a popular option owing to its mature nature as an established NIR platform, and due to the availability of a multitude of multi project wafer services that offer cheap access to sophisticated processes. However, due to the high material absorption of silicon dioxide in the 2.6–2.9 μm and beyond 3.6 μm wavelength ranges, the applications of SOI are limited. This has led to the development of modulators in new germanium based platforms with extended transparency ranges such as Germanium-on-Silicon (Ge-on-Si) and SiGe graded index—on silicon (SiGe-on-Si) [68]. Recently, intersubband transitions in III-V quantum wells have been exploited, as well as two dimensional materials such as black-phosphorus (BP) to demonstrate modulation in the mid-IR.

### 3.1. Modulation in Silicon Waveguides

The simplest method for achieving phase modulation in Si is to exploit the thermo-optic (TO) effect. A resistive heater is positioned close to a waveguide (but in the case of a metal heater, far enough away so that it does not introduce excess optical absorption), so that when a current is passed through it the waveguide is heated locally, and thus the waveguide’s effective refractive index ($\Delta n_{eff}$) is changed. The accompanying change in absorption coefficient (Δα) is negligible. In the mid-IR, the TO coefficient of Si (which relates the material’s temperature to its refractive index) reduces slightly compared to its value in the near-IR [69], but nevertheless remains strong. An SOI TO modulator was demonstrated at 3.8 μm in [70]. A TO phase shifter was placed in one arm of a Mach–Zehnder Interferometer (MZI), and was demonstrated to modulate light with an efficiency of 47 mW/π. It should be noted that, because in a phase shifter the change in phase is inversely proportional to wavelength, for an equivalent $\Delta n_{eff}$ the waveguide must be proportionally longer to achieve the same phase change, so the power consumption of a TO phase shifter would also increase proportionally. Furthermore, because the mode size also increases as wavelength is increased, metal heaters must be placed further away, reducing the overlap of any generated temperature increase and the optical mode, and further reducing the phase shifter efficiency. TO phase shifters operating at mid-IR wavelengths are thus intrinsically less power efficient than their direct near-IR equivalents (e.g., [71]). The key limitation of TO modulators is that their bandwidths are limited by slow thermal diffusion to the 10 s of kHz range, which is sufficient for some sensing and switching applications, but not for telecommunications.

Where larger bandwidths are required, the majority of reported short-wave infrared (SWIR) and mid-IR modulators using the SOI platform apply the free carrier plasma dispersion effect using carrier depletion or carrier injection diodes built into waveguides. The free-carrier plasma dispersion effect in silicon was investigated by Soref et al. [72] by analysing existing empirical spectral absorption coefficient data of highly doped Si wafers. This yielded the relationships between the electron/hole concentrations in Si and its absorption coefficient and refractive index, at the near-IR wavelengths of 1310 nm and 1550 nm. This approach was later extended by Nedeljkovic et al. [73] to long-wave IR (LWIR) wavelengths up to 14 μm, producing equations that can now be used to model the behaviour of Si free-carrier based EOMs operating in the SWIR and mid-IR.

High-speed silicon modulators optimised for the 2 μm band have been demonstrated. These typically employ MZIs, microring resonators (MRR) or Michelson interferometers [68]. The first demonstration of 2 μm modulation in Si was realised in a carrier-injection MZM with 1 mm long p-i-n phase shifters [74]. This device exhibited a switching speed of 1.5 GBaud, $V_{\pi }L_{\pi }$ of 0.12 V·mm and ER of 8.9 dB in 220 nm thick SOI platform. In the experimental measurement, the signal-to-noise ratio was low due to a lack of mid-IR amplifier and large IL due to close proximity of the high-doped regions and the optical mode. Further optimisation of the device geometry and amplification could improve the efficiency and bandwidth [74].

The first published carrier depletion modulator optimised for 2 μm was a MZM fabricated by the Cornerstone prototyping foundry in an SOI platform with a 220 nm top Si layer and 2 μm buried oxide (BOX) [75]. The MZM had two 2 mm long phase shifters, whose cross-section is shown in Figure 3. The modulator included travelling wave electrodes to ensure propagation of both RF and optical signals. The pn-junction was positioned in the middle of the waveguide, as in 1550 nm designs, but with an increased width to accommodate the larger mode and to increase confinement in the rib. High-doped region separation from the junction was optimised for lower loss while still providing interaction with the optical mode. The optical signal was amplified by a TDFA before being launched into the MZM. The first iteration of this MZM featured a cross-section that was optimised for operation at both 1.55 and 1.95 μm. Measured efficiencies were 2.02 V·cm and 2.68 V·cm at each wavelength, respectively. At 1550 nm, this device modulated at 30 GBaud with ER of 7.1 dB and at 1950 nm modulation speed was 20 GBaud with ER of 5.8 dB [75]. The second iteration of this device featured an updated phase shifter design optimised for modulation at 1950 nm. This modulator exhibited an efficiency $V_{\pi }L_{\pi }$ of 2.89 V·cm at a reverse bias of 4 V operating at 1.95 μm. The RF characterisation was performed by applying a pseudorandom on-off-keying (OOK) signal to each arm of the MZM with a peak amplitude of 2 V. At a wavelength of 1956.5 nm, the device modulated at 25 GBaud with ER of 6.25 dB. The IL was measured at 4.96 dB. The spectra, phase shift and insertion loss results are shown in Figure 4. The modulation speed in this experimental setup was limited by the detector bandwidth. The IL was predominantly in the phase shifter as reference structures with no metal layers were shown to have 4.76 dB IL. Michelson interferometer modulator (MIM) devices were also demonstrated on the same platform using looped mirrors as reflectors to double the efficiency compared to the traditional MZM design. A 1 mm long MIM exhibits around twice the efficiency compared to a MZM with the same length at $V_{\pi }L_{\pi }$ of 1.36 V·cm. This device modulated at 20 GBaud with ER of 1.97 dB. Degradation of signal beyond 20 GBaud was likely due to mismatch between optical and electrical signal propagation. To improve the bandwidth the length of the modulator should be reduced by employing a more efficient modulation method such as accumulation [2,8].

A 1.5 mm long MZM operating in the 2 μm window has been demonstrated with interleaved pn-junctions due to the advantages of mode overlap and misalignment tolerance. This device showed ER at DC conditions of >25 dB with IL of 3.2 dB. EO bandwidth was measured at 9.7 GHz [76]. Microring modulators in the same platform have been shown using a 15 μm radius all-pass configuration and employing a pn-junction offset toward the n${}^{+}$ region to optimise the reverse bias modulation at 1.97 μm. These modulators featured TiN heaters for thermal resonance tuning. They exhibited efficiencies of 2.2 to 2.6 V·cm at −1 V and −8 V reverse bias respectively. Open eyes were observed at 12.5 GBaud, which was limited by the commercial detector used in the experiment [77].

The electro-optic Kerr effect was used for modulators that can operate at cryogenic temperatures for applications in quantum technology [78]. The Kerr-effect has been explored in p-i-n phase shifter structures for modulation in the 2 μm band using multi-mode waveguides. Optical phase shift was measured at 0.09 rad/V. These devices greatly reduced optical loss due to minimised overlap between the free carrier regions and the optical mode loss was measured at 3.2 dB/cm [79] but at the cost of efficiency, footprint and drive voltages [80].

Recently the PAM-4 modulation format has become more popular as it has been chosen for the latest active IEEE Ethernet standard. This is a variation of the pulse-amplitude modulation format in which 4 levels represent data. Generating PAM-4 signals in the electrical domain is typically achieved by combining two non-return to zero (NRZ) signals with a digital-to-analogue converter (DAC). With increasing bandwidth this becomes difficult to design and is not cost effective, requiring highly linear drive amplifiers [81]. A viable alternative is to use MZM devices to generate PAM-4 signals in the optical domain. One such demonstrated solution is the use of a streamlined intensity-modulation, direct detection (IMDD) PAM-4 generation scheme.

This solution, shown in Figure 5 uses a dual-drive MZM in which each arm is driven by a different peak-to-peak power NRZ signal. In this scheme, one signal is amplified by 6 dB compared to the other. This enables the generation of four distinct intensity levels at the output of the MZM. Open eye diagrams were obtained at 12.5 GBaud with ER of 20 dB between the first and fourth levels shown in Figure 6. The high ER was likely due to the device not operating at quadrature. This could be controlled by employing a thermal phase shifter for a symmetrical MZM or by using an asymmetrical MZM and tunable laser source [8].

Modulation at 2 μm has also been shown in 220 nm SOI substrates using an MZM with serial PNP junctions in a back-to-back configuration. These modulators showed IL of 15 dB at peak transmission and efficiency of 1.6 V·cm at 8 V reverse bias. ER at DC conditions is 22 dB. Operated at RF, open eyes were obtained at 30 GBaud OOK with ER of 2.1 dB. When modulated by PAM-4 signal, this modulator was reported operating at 840 GBaud at a significantly decreased ER [9].

Currently the longest wavelength modulators fabricated on SOI substrate were shown operating at the edge of the SiO${}_{2}$ transparency range—3.8 μm [82]. PIN diodes were used as phase shifters to inject carriers in an MZI by using the free-carrier electrorefraction effect and in a variable optical attenuator (VOA) by using free-carrier electroabsorption effect. The cross section of the diode used in both configurations is shown in Figure 7.

The VOA modulated at DC to a depth of 34 dB and at AC rated up to 60 MBaud. The electrorefraction modulator(ERM) reached ER of 22.2 dB at DC with $V_{\pi }L_{\pi }$ of 0.52 V·cm and AC rate of 125 MBaud [82]. The transmission spectra is shown in Figure 8. The bandwidth governing parameter of this device was the separation between the Ohmic contact regions from the waveguide core. When this separation is small then the high doped regions introduce excess loss due to overlap with the optical mode. Since optical mode size scales with wavelength for these devices Ohmic contacts need to be placed further away, this increases the volume of the intrinsic region of the device as well as diode junction resistance which limits modulation efficiency and bandwidth.

BP is a two-dimensional (2D) material which shows potential for active functions in mid-IR devices. A hybrid system of 2D BP and Si photonics has been used to realise a waveguide integrated EOM operating in the spectral range from 3.85 μm to 4.1 μm [83]. The modulator was made up of indium tin oxide (ITO)-Al${}_{2}$O${}_{3}$ (aluminium oxide)-BP capacitor on top of a Si waveguide. BP flakes were exfoliated by tape and then oriented along the TE direction of the waveguide using a micromanipulator under a microscope. This fabrication method may face difficulty with wafer scaling compared to conventional mid-IR platforms. BP exhibits absorption change when bias voltage is applied to it, which introduces an intensity modulation in the mode travelling along the waveguide. The absorption of BP shows steep roll off after 3.5 μm and is cut off by 4.2 μm, which limits the wavelength of this type of device. The modulator was demonstrated operating under modulation frequency of 10 kHz where ER ≈ 5 dB is achieved at 3.85 μm [83]. This device has a smaller footprint of 225 μm${}^{2}$ and lower switching energy 2.7 pJ compared to the traditional mid-IR platforms.

Lithium niobate (LiNbO${}_{3}$) is transparent at wavelengths up to 5 μm with a high electro-optic coefficient. Recent work on integrating crystalline silicon with LiNbO${}_{3}$ through wafer bonding and thin-film transfer techniques has enabled the development of a new platform that exploits the Pockels effect in the mid-IR. The waveguide was made in the Si layer; however, the mode resided mainly in the LiNbO${}_{3}$ area (56% mode overlap). MZMs have been demonstrated working with ER of ≈8 dB, $V_{\pi }L_{\pi }$ of 26 V·cm and IL of 3.3 dB operating at 3.39 μm [84].

Although silicon is transparent up to 8 μm wavelength, the SiO${}_{2}$ in the common SOI platform limits its application for wavelengths longer than 4 μm. Suspended Si waveguides have been shown with low loss at up to 7.7 μm wavelength [85]; however, modulators have not yet been demonstrated in this platform, and the fabrication of such suspended active devices may be challenging. Alternative waveguide core materials must therefore be considered to reach longer wavelengths.

### 3.2. Modulation in Germanium Waveguides

Germanium is recognised as having an excellent potential for mid-IR photonics owing to its wide transparency, which covers a range from 2 μm to 16 μm [68], and low loss waveguides have been demonstrated in the Ge-on-Si platform at wavelengths up to 11 μm [86], in SiGe-on-Si at up to 11.2 μm [87], and in suspended germanium up to 7.7 μm [88,89]. However, these waveguiding demonstrations have been relatively recent, and so Ge and SiGe-on-Si modulators are also at a very early stage of development.

The thermo-optic effect is stronger in Ge than in Si, and remains strong throughout the mid-IR [69]; for example, at 3.5 μm wavelength the TO coeffiecient of Ge is 2.5 times greater than that of Si [69]. Fujigaki et al. demonstrated a TO phase shifter in a Germanium-on-insulator (GOI) waveguide platform at 1.95 μm [90] with a very low 7.8 mW/π efficiency, which was helped not only by Ge’s strong TO coefficient, but also by the GOI platform’s high refractive index contrast and resulting high mode confinement. However, at longer wavelengths the buried oxide layer in GOI is not transparent (just as in SOI), so the platform cannot be used. TO phase shifters have been investigated in GOS and Ge-on-SOI waveguides in [91]. The GOS phase shifters had a high power consumption of 350 mW/π, because the platform suffers from high thermal conductivity of the Si substrate layer, which allows the thermal power to rapidly be conducted away from the waveguide. An additional challenge with TO phase shifters in the GOS platform is that there is no conventional transparent top cladding layer that is used to separate the waveguide from the heaters placed directly above it, in the way that SiO${}_{2}$ is routinely used with SOI waveguides. Instead in [91] the metal heater was placed on a Ge pedestal positioned a few micrometers to the side of the waveguide core. The efficiency was improved by moving the waveguides to the Ge-on-SOI platform, in which there is a buried oxide layer within the Si substrate that is placed far enough below the Ge waveguide core so as to isolate the optical mode from the buried oxide layer, but the waveguide dimensions are otherwise the same. The efficiency was 56 mW/π thanks to the much smaller thermal conductivity of the oxide layer, and was further reduced to 8 mW/π by fully suspending the device by oxide removal. A first demonstration of SiGe-on-Si TO phase shifters was shown in [92] with a low efficiency; however, the authors state that the device geometry had not yet been optimised for high efficiency. Nevertheless, the platform would suffer from the same thermal conductance issues as the GOS platform.

As with Si, the free carrier effect is required to reach high modulation bandwidths in Ge. The free carrier absorption (FCA) effect in Ge has been numerically predicted in [93], partly based on experimental absorption coefficient data for doped Ge from various literature sources, and partly based on first-principles theoretical calculations. The work gives equations for estimating Δn and Δα in Ge due to free carriers at wavelengths between 2 μm and 16 μm. In general, the prediction is that holes are more absorbing than electrons in Ge, and that Δα for both electrons and holes is larger than in Si for all wavelengths above 3 μm. Overall, then, absorption modulators are expected to be more efficient in Ge than in Si; however, phase shifters are expected to suffer from high accompanying parasitic absorption, particularly at longer wavelengths.

FCA modulation in germanium has been demonstrated in a variable optical attenuator in the germanium-on-insulator platform operating at 1.95 μm [94]. By introducing phosphorus spin-on-glass doping technology, which provides steep purity profile and low defect density, Ge VOAs have shown modulation efficiencies of 360–380 dB/A. This device shows a comparable modulation efficiency to the SOI VOA operating at 3.8 μm [82].

Ge-on-Si modulators have been demonstrated operating at 3.8 μm and 8 μm [10]. They were based on PIN diodes integrated with Ge rib waveguides with 3 μm core thickness (T${}_{rib}$). Two different types of modulator were demonstrated in this platform: VOAs and MZMs. The cross-section of the diode is shown in Figure 9. VOA devices were also investigated at 8 μm wavelength with a diode length of 2 mm.

The 1 mm long electro-absorption modulators exhibited ER of 35 dB at DC with 7 V of forward bias. A similar 1 mm long MZM had $V_{\pi }L_{\pi }$ of 0.47 V·cm and over 13 dB of ER. Both modulators showed open eye diagrams at 60 MBaud while at 2.5 V and 1.9 V driving voltages, respectively. The spectra and phase shift versus current of the MZM are shown in Figure 10. VOA devices operating at 8 μm demonstrated 2.5 dB ER at 7 V DC forward bias voltage. Measurements showed that injected carrier absorption coefficient was 4.9 times greater at 8 μm compared to 3.8 μm at the same injection current (per unit length) shown in Figure 11. The low modulation efficiencies could be explained by a combination of factors, including overly large Ohmic contact separation, high Ohmic contact resistance and high defect concentration at the Si-Ge interface (which may cause a short carrier lifetime). An improvement to efficiency could be achieved by reducing the contact separation, which would improve the overlap of the injected carrier concentration with the optical mode, without introducing a lot of excess optical loss [10].

Free-carrier modulation has also been demonstrated in a SiGe-on-Si platform to enable modulators operating at a wide wavelength range spanning from 6.4 μm to 10.7 μm [11]. These devices rely on a graded index platform in which Ge concentration linearly increases from Si to Ge through a 6 μm thick layer, which is on top of a 500 μm thick Si substrate. The diode is created vertically with metal contacts placed on the back of the substrate and on top of the waveguide. This device consists of a straight waveguide and thus exploits absorption modulation.

A 2.6-mm-long modulator is estimated to have IL of 4.2, 8.0 and 15.6 dB at 6.4, 8.5 and 10.7 μm respectively. A maximum ER of 1.3 dB is obtained at 10.7 μm which is to date the largest electro-optical modulation at wavelengths longer that 10 μm in group IV waveguides. Modulation is achieved at 30 MBaud while operating the device in the injection regime with bias voltage of 1 V, and 150 MBaud operating in depletion regime with 4 V bias. The static transmission relative to zero bias voltage applied is 0.45 dB at 8 V bias [11].

In Table 2, we provide a summary of the demonstrated performances of the abovementioned mid-IR modulators.

## 4. Discussion

At the shorter wavelength end of the mid-IR spectrum, the plasma dispersion effect in silicon is moderately enhanced for both the refractive index change and the absorption change, but is still dominated by the refractive index change. Therefore, for the 2 μm wavelength band, it is possible to implement a similar design principle as conventional O-band and C-band modulators to realise high-speed operation. The carrier distribution in the PN junction under various bias conditions is independent from the optical wavelength. However, the longer optical wavelength implies a larger optical mode and wider single mode waveguide. This leads to two challenges: the larger mode will overlap the depletion region more and this interaction often weakens the efficiency. Optimisation of the position of the highly doped region and metal contact is therefore required to reduce optical loss, and this may cause reduced high-speed performance due to longer access length. Therefore, the design of a 2 μm wavelength modulator is more than a reuse from a C-band modulator design; careful optical modelling, carrier distribution modelling and AC simulation are required to ensure an appropriate performance.

Apart from phase shifter cross-section optimisation, plasma dispersion modulators operating at 2 μm could benefit from implementing strategies similar to their near-IR counterparts such as WDM by developing AMMIs operating at this wavelengh. Another direction worth exploring would be flip-chip bonding of an integrated driver similar to those demonstrated in [30]. Only recently, there have been explorations of different coding schemes such as PAM-4 [8]. This could further be investigated by incorporating new MZM architectures such as a dual-parallel or a segmented MZM. Finally germanium based platforms could be explored for phase shift modulation, given it becomes transparent in the 2 μm range and it exhibits higher absorption change and similar refractive index change at 2 μm for the same carrier concentration compared to silicon [93]. This would indicate that germanium modulators relying on electroabsorption would be likely candidates to compete with silicon-based ones. GOI VOAs have already been demonstrated showing comparable performance to SOI attenuators in the MIR.

Towards longer wavelengths, the absorption change becomes more prominent relative to the refractive index change in both Si and Ge. Phase change is inversely related to wavelength in a MZI device. At longer wavelengths phase change diminishes so devices need to be longer which increases absorption losses. EAMs are widely adopted at these longer wavelengths. The carrier injection type is often implemented as it can offer larger absolute absorption change when high speed is not the priority for applications at these wavelengths, whereas carrier depletion devices will suffer from increased losses due to large mode overlap with background doping regions.

On the other hand, germanium and silicon-germanium-based platforms extend the operating window of active devices further into the MIR due to Ge’s wider transparency region of up to 16 μm. Designing modulators for longer wavelengths becomes challenging as the optical mode size increases and its overlap with phase shifter region diminishes. This leads to lower modulation efficiencies as well as higher optical loss due to the high doping layer requirements of modulators operating on the free carrier effect. Placing those layers further away from the optical mode would then limit high speed performance by increasing access resistance in the electrical signal path. To date, modulators extending the furthest into the MIR reported make use of a graded structure which allows for broadband mode confinement on top of the waveguide and reduces mode overlap with highly doped layers. However, these devices are operating in the carrier injection regime, for which the minority carrier lifetime governs the bandwidth. As such, reported modulation speeds are two orders of magnitude lower compared to depletion devices operating at lower wavelengths.

## 5. Conclusions

Near-IR silicon photonics modulators, especially the all-silicon ones, have been demonstrated with high bandwidth beyond 100 GBaud PAM-4 for both MRMs and MZMs. The MOSCAPs show a good compromise between bandwidth and footprint for current 50 GBaud PAM-4 operation, and the carrier injection type modulators have been demonstrated higher bandwidth when integrated with equalisers. All these modulators have their benefits in terms of specific scopes such as footprints, high bandwidth and low loss. The development of co-designed modulators and drivers shows that all silicon modulators can meet the bandwidth and power consumption requirements for the photonics transceiver applications [4,30]. The success of these paves the way for the development of the mid-IR modulators.

We have presented published works on group IV modulators in the mid-IR field operating at wavelengths from 1.95 μm to 10.7 μm. As work on optimising silicon-on-insulator MZMs has matured, new alternatives for pushing the field forward have emerged. Towards shorter wavelengths in the SWIR, we have shown that PAM-4 modulation in the digital domain allows for higher spectral efficiency compared to OOK while retaining traditional single carrier depletion MZM designs. Our work has achieved data rates of 12.5 GBaud with comparable IL, ER and device efficiency to OOK MZM designs [8]. Works employing DSP techniques such as FFE have been able to reach rates of 40 GBaud [9]. We suggest that future work in the 2 μm band could benefit from following recent trends in NIR modulators such as WDM, integration, packaging of photonics and electronics as well as design of optical domain PAM devices to reduce power draw of complicated DSP generated modulation schemes.

Towards longer wavelengths, high losses present in silicon dioxide above 4 μm limit the operation of SOI active devices. Germanium has shown to be the most promising group IV material for a host of MIR applications as it exhibits higher carrier mobility, nonlinear coefficients and a larger transparency window than silicon. In this paper, we have presented recent works on active devices based on germanium waveguides for the MIR. Our work has achieved modulation speeds of 60 MBaud with DC ER of 35 dB at 3.8 μm [82] and we have demonstrated EAMs operating at up to 8 μm [82]. Additionally, graded GeSi structures have been demonstrated operating at up to 10.7 μm at 225 MBaud speed but at the expense of a high insertion loss [9]. Finally, we have shown recent works on new hybrid platforms such as Si-BP [83] and Si-LiNbO${}_{3}$ [84] exhibiting high efficiencies with lower modulation speeds and fabrication scaling difficulties. Future work in the LWIR spectrum will involve further optimisation of phase shifter designs to improve their performance.

## Figures and Tables

**Figure 1 sensors-22-09620-f001:**
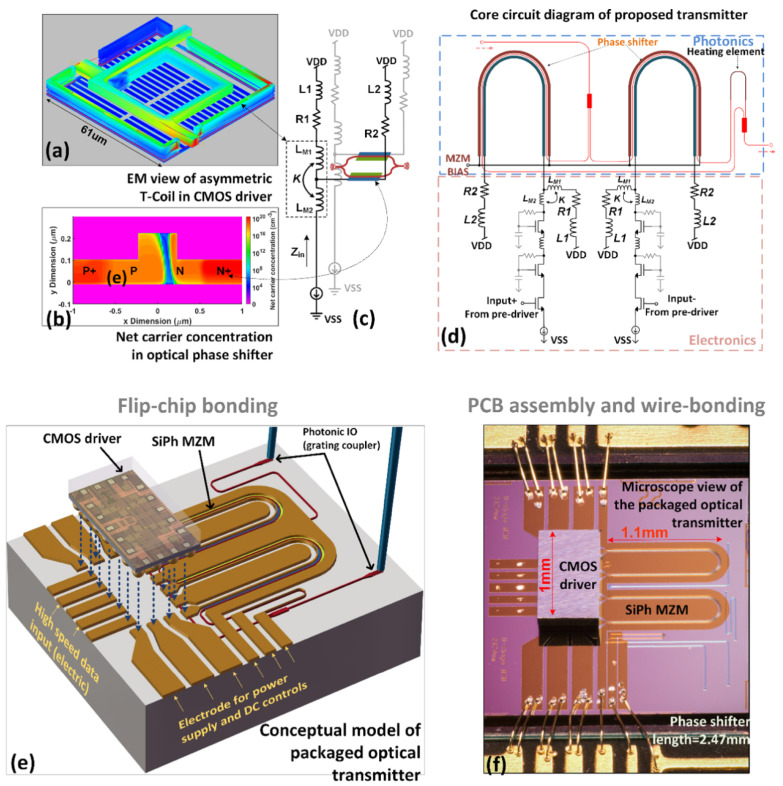
Illustration of the convergence of a CMOS driver and silicon photonics modulator [53]. (**a**) EM view of asymmetric T−coil. (**b**) Net carrier concentration in silicon photonics carrier depletion phase shifter. (**c**) The input impedance of T−Coil with a travelling wave phase modulator considered as load. (**d**) Core circuit diagram of the device. (**e**) Conceptual model of the packaged optical transmitter. (**f**) Microscope view of the packaged optical transmitter.

**Figure 2 sensors-22-09620-f002:**
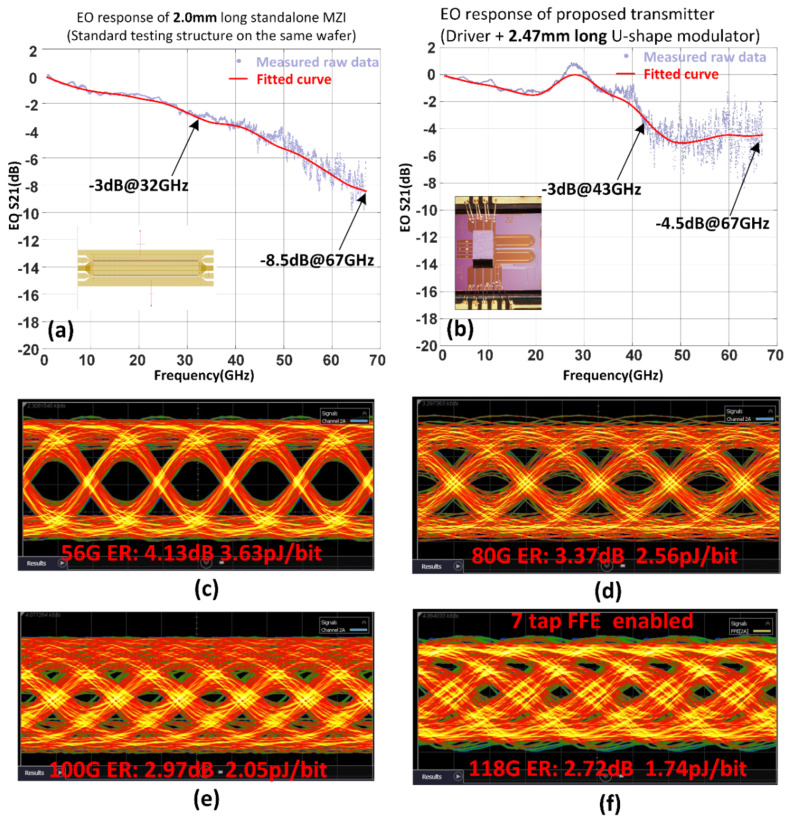
Measurementresults of the 100 GBaud optical transmitter [53]. (**a**) Measured EO response of 2.00 mm long standalone MZM (built for comparison). (**b**) Measured EO response of the proposed all silicon transmitter (**c**) Measured eye-diagram at 56 G OOK (**d**) Measured eye-diagram at 80 G OOK (**e**) Measured eye-diagram at 100 G OOK (**f**) Measured eye-diagram at 118 G OOK with 7 tap FFE enabled on DCA.

**Figure 3 sensors-22-09620-f003:**
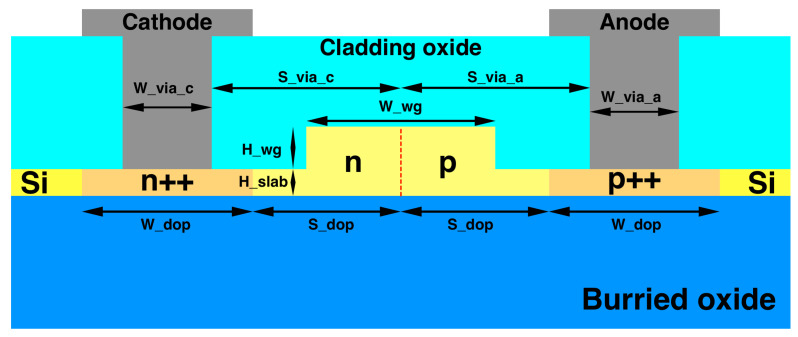
Phase shifter cross-section of mid-IR depletion modulator [8].

**Figure 4 sensors-22-09620-f004:**
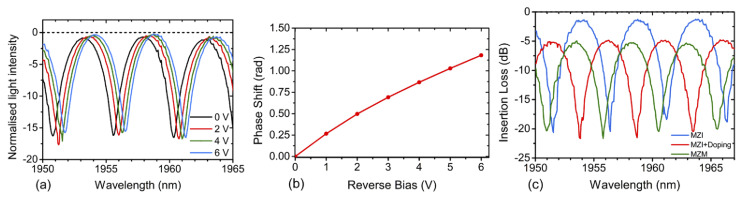
(**a**) Normalisedlight intensity spectra from the MZM at 0–6 V reverse bias voltages. (**b**) Phase shift for a MZI modulator with 2 mm long phase shifter. (**c**) Insertion loss of the 2 mm MZM device and its two reference structures (MZM without metal electrode; MZI without metal nor doping) [75].

**Figure 5 sensors-22-09620-f005:**
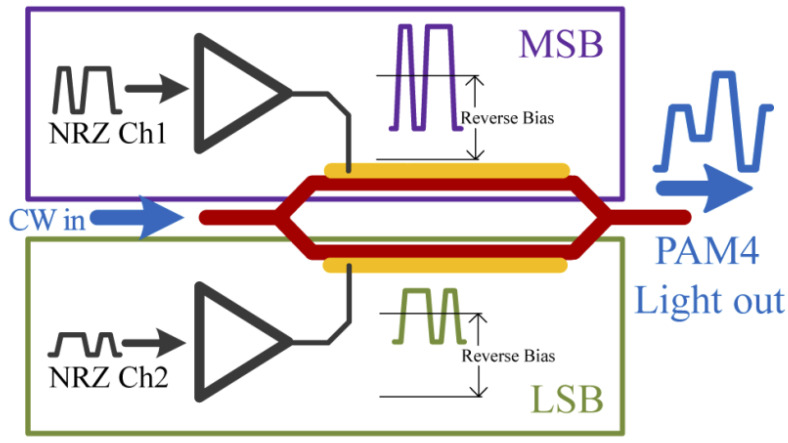
Streamlined PAM-4 modulation using a dual-drive MZM. Upper arm signal is amplified to be 6dB larger than the lower arm signal [8].

**Figure 6 sensors-22-09620-f006:**
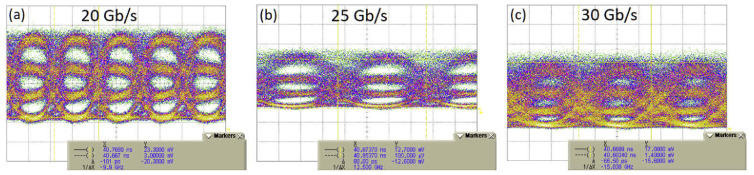
Eye diagram showing PAM-4 operation of a 2 mm MZI modulator at a wavelength of 1950 nm. (**a**) 10 GBaud PAM-4. (**b**) 12.5 GBaud PAM-4. (**c**) 15 GBaud PAM-4 [8].

**Figure 7 sensors-22-09620-f007:**
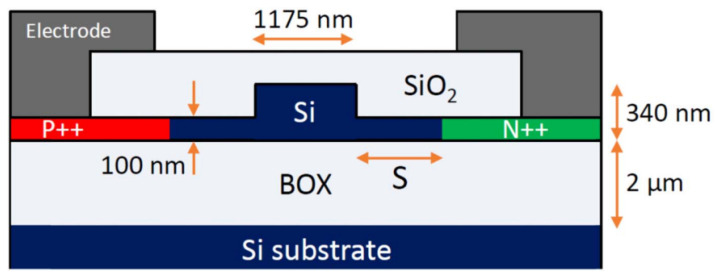
Schematic cross section of SOI injection modulators from [82].

**Figure 8 sensors-22-09620-f008:**
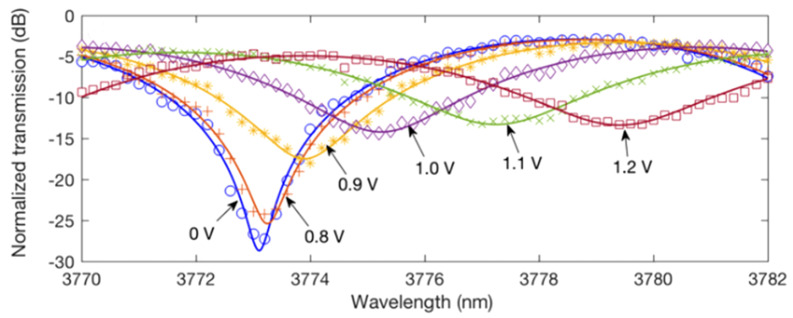
Normalisedtransmission spectra of the MZM operating at 3.8 μm for varying applied forward bias voltages from [82].

**Figure 9 sensors-22-09620-f009:**
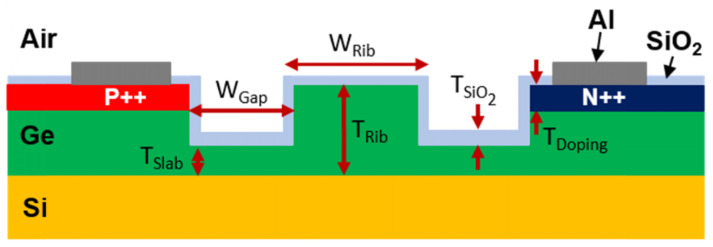
Schematic cross section of Ge-on-Si PIN diodes for injection modulators [10].

**Figure 10 sensors-22-09620-f010:**
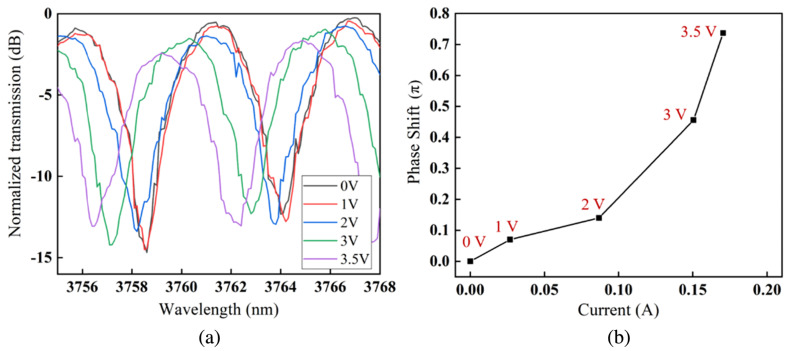
(**a**) Optical spectra of the GOS MZM under several DC voltages. (**b**) Phase shift versus current of the MZM from [10].

**Figure 11 sensors-22-09620-f011:**
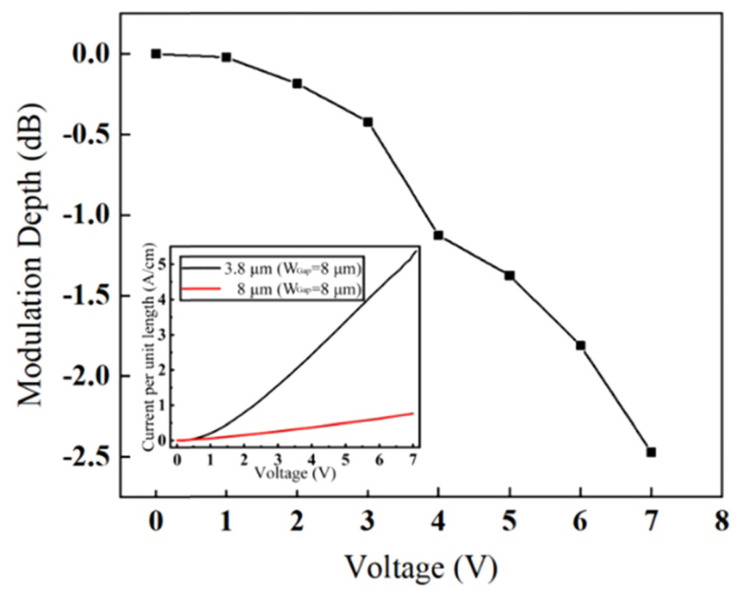
Modulation depth versus applied voltage for a GOS EAM operating at a wavelength of 8 μm. Inset: current per unit length versus voltage at wavelengths of 8 μm and 3.8 μm from [10].

**Table 1 sensors-22-09620-t001:** State-of-the-art device performance for different types of silicon modulators in the NIR spectrum.

Work Ref.	Material Platform	Type	Modulation Method	Power Efficiency	Modulation Speed
[4]	SOI	MZM	Depletion	0.54 pJ/bit	112 GBaud ^1^
[54]	SOI	MRM	Depletion	-	120 GBaud ^1^
[29]	SOI	MZM	Accumulation	2.4 pJ/bit	50 GBaud ^1^
[30]	SOI	MRM	Accumulation	-	100 GBaud ^2^
[55]	SOI	MZM	Injection	1.2 pJ/bit	70 GBaud ^1^

^1^ PAM-4 modulation scheme ^2^ OOK modulation scheme.

**Table 2 sensors-22-09620-t002:** Comparison of modulators operating in the mid-IR.

Work Ref.	Material Platform	Type	λ	IL	ER	Efficiency	Modulation Speed
[8]	SOI	MZM	1.95 μm	4.96 dB	6.25 dB	2.89 V·cm	25 GBaud ^2^
[8]	SOI	MIM	1.95 μm	4.1 dB	1.97 dB	1.36 V·cm	20 GBaud ^2^
[76]	SOI	MZM	1.95 μm	3.2 dB	>25 dB ^1^	-	12.5 GBaud ^2^
[77]	SOI	MRM	1.97 μm	10–20 dB	20.1 dB ^1^	2.2–2.6 V·cm	12.5 GBaud ^2^
[8]	SOI	MZM	1.95 μm	4.96 dB	20 dB ^1^	2.89 V·cm	12.5 GBaud (PAM-4) ^2^
[9]	SOI	MZM	1.95 μm	15 dB	2.1 dB (@15 GBaud)	1.6 V·cm	40 GBaud (PAM-4) ^3^
[95]	SOI	MRM	1.95 μm	8 dB	1.92 dB (@35 GBaud)	0.85 V·cm	50 GBaud ^3^
[82]	SOI	VOA	3.8 μm	4.96 dB	34 dB ^1^	-	60 MBaud
[82]	SOI	ERM	3.8 μm	4.96 dB	22.2 dB ^1^	0.52 V·cm	125 MBaud
[83]	Integrated Si and BP	Waveguide EAM	3.85 μm	-	5 dB	-	30 kHz
[84]	Integrated Si and LiNbO${}_{3}$	MZM	3.39 μm	3.3 dB	Data	26 V·cm	23 kBaud
[10]	Ge on Si	Waveguide EAM	3.8 μm	-	>35 dB ^1^	-	60 MBaud
[10]	Ge on Si	MZM	3.8 μm	-	13 dB	0.47 V·cm	60 MBaud
[10]	Ge on Si	Waveguide EAM	8 μm	-	2.5 dB	-	-
[11]	Graded SiGe	Waveguide EAM	10.7 μm	15.6 dB	1.3 dB	-	225 MBaud
[94]	Ge on insulator	VOA	1.95 μm	-	20 dB	380–460 dB/A	-

^1^ This ER is obtained at DC conditions. ^2^ Tested with a 12.5 GHz-bandwidth PD. ^3^ Tested with a 22 GHz-bandwidth PD, with root raised cosine filter and feed-forward equalisation (FFE).

## Data Availability

Data sharing not applicable no new data were created or analyzed in this study. Data sharing is not applicable to this article.

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
