# Peer review of "Near-IR & Mid-IR Silicon Photonics Modulators"

_sensors, 2022, doi:10.3390/s22249620_

Round 1
Reviewer 1 Report
The submitted paper is devoted to the well-known problems related to the transmission parameters of photonic circuits operating in the near and mid-infrared range. The authors of the paper performed a synthetic classification and analysis of integrated photonics. The state-of-the-art analysis has been conducted quite extensively. The reviewer believes that the paper can be published after some corrections and extensions have been made:
1.The abstract must be significantly extended, because in the presented form it does not contain anything about the conducted research and the results obtained.
2.Line 22 - the modulation rate should be specified in bauds. The entire text of the document should be reviewed and the method of assessing the maximum bandwidth of individual photonic circuit should be standardized.
3.The purpose and motivation of the analyzes carried out should be clearly specified and the place and manner of using the obtained results should be presented.
4.It is worth presenting the transfer characteristics of the analyzed integrated photonics/modulators on graphs, which will be much more useful than text descriptions. Good examples are given in Figure 2.
5.Not all abbreviations used in the text are expanded and included in the abbreviations list. The list of abbreviations should be in alphabetical order.
6.An additional chapter should be written in which the application of classified integrated photonics systems will be broadly defined, broken down into areas, e.g. sensorics, fiber optic and wireless telecommunications, medicine, and telemetry. In the further part of the paper, it is necessary to refer to these descriptions based on the requirements set in individual fields.
7.The paper should be marked as "Review" in the header.
Reviewer 2 Report
Manuscript ID: sensors-2026669
Review of the manuscript “Near-IR & Mid-IR silicon photonics modulators” by G. Georgiev et al. submitted for consideration to Sensors.
In this manuscript, the authors have discussed the current progresses of the near-IR and mid-IR group IV silicon photonic modulators from the early fabricated silicon modulators to very recent research in this field of study where they have mentioned hybrid platforms. For shorter wavelengths in the SWIR and longer wavelengths in the LWIR, facing challenges and prospects have been summarized. I believe the manuscript is original and it has been scientifically written. This manuscript is what it is expected from a topical review article as it can lead the readers to obtain appropriate information about the research topic of the near-IR and mid-IR silicon photonic modulators.
Therefore, I do support the publication of the manuscript. However, I will bring out some points to be clarified as part of the changes for the manuscript.
- I recommend that the authors add some sentences to discuss whether or not the “Two photon absorption” has any effect on the performance of the mentioned modulators, specifically at near-infrared wavelengths.
- The caption for fig. 1(f) was missed. Please correct it.
- Figure 2(f) was not referred in the text. It seems it must be mentioned somewhere in the lines 158-160.
Round 2
Reviewer 1 Report
Almost all the recommended corrections indicated in the previous version of the review were taken into account. Thank you for introducing corrections and extensions to the paper in accordance with the reviewer's recommendations. I have no further comments.